# Cesarean Scar Thickness Decreases during Pregnancy: A Prospective Longitudinal Study

**DOI:** 10.3390/medicina58030407

**Published:** 2022-03-09

**Authors:** Egle Savukyne, Egle Machtejeviene, Mindaugas Kliucinskas, Saulius Paskauskas

**Affiliations:** Department of Obstetrics and Gynecology, Medicine Academy, Lithuanian University of Health Sciences, 44307 Kaunas, Lithuania; egle.machtejeviene@lsmuni.lt (E.M.); mindaugas.kliucinskas@lsmuni.lt (M.K.); saulius.paskauskas@lsmu.lt (S.P.)

**Keywords:** cesarean section scar, scar thickness, transvaginal ultrasonography

## Abstract

*Background and Objectives:* The aim of this study is to evaluate changes in uterine scar thickness after previous cesarean delivery longitudinally during pregnancy, and to correlate cesarean section (CS) scar myometrial thickness in the first trimester in two participants groups (CS scar with a niche and CS scar without a niche) with the low uterine segment (LUS) myometrial thickness changes between the second and third trimesters. *Materials and Methods:* In this prospective longitudinal study, pregnant women aged 18–41 years after at least one previous CS were included. Transvaginal sonography (TVS) was used to examine uterine scars after CS at 11–14 weeks. The CS scar niche (“defect”) was defined as an indentation at the site of the CS scar with a depth of at least 2 mm in the sagittal plane. Scar myometrial thickness was measured, and scars were classified subjectively as a scar with a niche (niche group) or without a niche (non-niche group). In the CS scar niche group, RMT (distance from the serosal surface of the uterus to the apex of the niche) was measured and presented as CS scar myometrial thickness in the first trimester. The myometrial thickness at the internal cervical os was measured in the non-niche group. The full LUS and myometrial LUS thickness at 18–20 and 32–35 weeks of gestation were measured in the thinnest part of the scar area using TVS. Friedman’s ANOVA test was used to analyse scar thickness during pregnancy and Mann–Whitney test to compare scar changes between CS scar niche and non-niche women groups. For a pairwise comparison in CS scar thickness measurements in the second and third trimesters, we used Wilcoxon Signed Ranks test. *Results:* A total of 122 eligible participants were recruited to the study during the first trimester of pregnancy. The scar niche was visible in 40.2% of cases. Uterine scar myometrial thickness decreases during pregnancy from 9.9 (IQR, 5.0–12.9) at the first trimester to 2.1 (IQR, 1.7–2.7) at the third trimester of pregnancy in the study population (*p* = 0.001). The myometrial CS scar thickness in the first trimester (over the niche) was thinner in the women’s group with CS scar niche compared with the non-niche group (at internal cervical os) (*p* < 0.001). The median difference between measurements in the CS scar niche group and non-niche group between the second and third trimester was 2.4 (IQR, 0.8–3.4) and 1.1 (IQR, 0.2–2.6) (*p* = 0.019), respectively. Myometrial LUS thickness as percentage decreases significantly between the second and third trimester in the CS scar niche group compared to the non-niche group (U = 1225; z = −2.438; *p* = 0.015). *Conclusions:* CS scar myometrial thickness changes throughout pregnancy and the appearance of the CS scar niche was associated with a more significant decrease in LUS myometrial thickness between the second and third trimesters.

## 1. Introduction

Cesarean delivery rates have risen dramatically in recent decades as cesarean section (CS) has become the most performed obstetric procedure worldwide [1]. Uterine scar defect appears to be a frequent complication of CS. It may result in many obstetric complications such as ectopic scar pregnancy, placenta accreta, uterine scar dehiscence, or rupture [2]. It was suggested that residual myometrial thickness (RMT) over the CS scar niche in the first trimester of pregnancy could predict uterine scar dehiscence or rupture at vaginal birth after CS. Moreover, the measurement of RMT (from the apex of the uterine scar niche to the serosa) by transvaginal ultrasonography may become a valuable tool to predict obstetric complications in a subsequent pregnancy and delivery [3,4]. Thin low uterine segment (LUS) thickness, as measured by ultrasound examination in the second and third trimester of pregnancy, was associated with a potential risk of uterine scar dehiscence and rupture during a trial of vaginal delivery after CS [5,6,7]. Unfortunately, there is no clear correlation between CS scar thickness over the niche at the first trimester of pregnancy, as CS scar niche appears in the first trimester and myometrial LUS thickness appears in the second and third trimesters. In this prospective longitudinal study, we aim to evaluate changes in CS scar thickness during pregnancy in CS scar niche and non-niche groups and investigate how uterine scar niche can affect the feasibility of a decrease in myometrial LUS thickness between the second and third trimesters. Yet, this study was not powered to answer the question of a possible association between thinner CS scar at the second and third trimester and uterine rupture or dehiscence in a subsequent pregnancy.

## 2. Materials and Methods

A prospective longitudinal study was performed between March 2019 and October 2020 at the Department of Obstetrics and Gynecology of Lithuanian University of Health Sciences, Kauno Klinikos, a high-risk pregnancy referral center, with 3000 deliveries every year. The Kaunas regional bioethics committee approved the study (protocol No. BE-10-15). The trial registration date was 18 March 2019 (Australian New Zealand Clinical Trials Registry, registration No. ACRN12619000435189).

Women over 18 with a singleton pregnancy after at least one previous low-transverse CS and at least twelve months after the last CS were recruited at the 11^+0^–13^+6^ week ultrasound screening after the procedures had been explained to them fully. Written informed consent was obtained from all participants and transvaginal ultrasound was performed for a uterine scar examination using a Voluson E8 Expert device (GE Healthcare Ultrasound Korea, Ltd., Seongnam-Si, Korea) with a 4–9 MHz probe. The standardised plane for a CS scar evaluation was performed as a midsagittal view of the uterine isthmus and internal cervical os. CS scar niche was defined as an anechoic area with at least 2 mm depth at the scar site. According to sonographic findings, the study participants were assigned subjectively to the CS scar niche group (niche group) or to the CS scar without niche group (non-niche group). In the CS scar niche group, RMT (distance from the serosal surface of the uterus to the apex of the niche) was measured and presented as CS scar myometrial thickness in the first trimester (Figure 1). The myometrial thickness at the internal cervical os was measured in the non-niche group (Figure 1). Participants needed to have their bladder empty during transvaginal sonography (TVS) in the first trimester. Demographic and obstetric history was obtained during the first scan visit. Second ultrasound examination at 18^+0^–20^+0^ weeks and a third at 32^+0^–35^+6^ weeks were planned. TVS on a full bladder was used to measure the full LUS and myometrial thickness to delineate the scar area clearly, as was previously described [5] (Figure 2). LUS is a two-layer structure with a hyperechogenic layer, including the bladder wall and a hypoechogenic layer that is considered to represent the myometrium [5] (Figure 2). At least three measurements were taken, with the lowest value being retained. The same investigator was the first researcher (S.E.) and performed all scans as the same ultrasound examination method was applied for CS scar evaluation. All representative images were stored on the local hospital image storage system (MEDDREAM-172.27.1.146). Data were recorded prospectively on an SPSS spreadsheet. Data about pregnancy outcomes and complications were retrieved from an electronic hospital database.

The median and 25th–75th percentile for RMT, myometrial thickness near the internal os at the first trimester, and full LUS as myometrial LUS thickness was calculated for the second and third trimester of pregnancy. These characteristics were compared between the first, second, and third trimesters using two-way Friedman’s ANOVA test. For paired samples, to compare LUS myometrial thickness at the second and third trimester in the CS scar niche group and non-niche group, we used Wilcoxon Signed Rank test. The change in percentage was calculated using Mann–Whitney test (scar myometrial thickness at the third trimester—scar myometrial thickness at second trimester/myometrial thickness at the second trimester × 100%. The more negative result—the more significant decrease in CS scar thickness).

The statistical significance of the difference in categorical data was determined using the Chi-square test or Fisher’s exact test. Continuous variables were compared between groups (CS scar niche group versus non-niche group) using the Mann–Whitney U-test. *p* < 0.05 was considered statistically significant.

## 3. Results

A total of 122 eligible participants were included in the study at the first trimester of pregnancy. Sonographic data were obtained from 120 participants who attended scans at the first and second trimesters and 117 of these attended all three scans (Figure 3). Complete sonographic data were obtained from 117 participants who attended all scans during pregnancy. 

The CS scar was visible in 95 (77.9%) cases. CS scar niche was found in 49 women (40.2%), as the non-niche group consisted of 73 (59.8%) women. A total of 94 (77.4%) women had undergone one previous CS, 26 had undergone two CS (21.3%), and 2 patients had undergone three CS (1.6%). CS scar niche was visible in 83.7% of women after one previous CS and 16.3% cases after two or more previous CS (*p* = 0.228). Five (4.0%) participants had a first-trimester RMT below 2.0 mm, two after the last two CS, and three after the previous one CS. The mean gestational age in the study population at delivery was 38.8 ± 2.37 weeks with a neonatal weight of 3473.7 ± 598.0 g. Outcomes of the study population are shown in Table 1.

Of the 63 women who underwent a trial of labor after one previous CS, 41 (65.0%) had a successful vaginal delivery. In the patient group with CS scar niches (*n* = 49) compared to the non-niche group (*n* = 46), there were no statistical differences in the type of delivery. A total of 19 women had successful trials of labor in the niche group and 22 in the non-niche group (38.7% vs. 47.8%, *p* = 0.802).The median neonatal weight did not differ between women groups according to the CS scar niche as Apgar score. No significant association was found between CS niche and maternal age, BMI, gestational diabetes, smoking status, and previous vaginal birth (Table 2). Demographic and main obstetric history of the study participants in the CS scar niche group and non-niche group is shown in Table 2. Overall, the median CS scar myometrial layer thickness in the study population (CS scar niche group and non-niche group) reduced from 9.9 mm during the first trimester to 2.1 mm in the third trimester of pregnancy (*p* < 0.001) (Figure 4).

The decrease in CS scar myometrial thickness between the first and second trimesters was higher throughout the study population compared with a period between the second and third trimester (8.9 vs. 1.8 mm *p* < 0.001). The median full LUS thickness (one caliper placed at the interface between amniotic fluid and the deciduas) in the second trimester was 6.8 mm and 4.0 mm in the third trimester. The change in the full LUS thickness between the second and third trimesters was significant at 2.5 (1.2–4.9) mm (*p* < 0.001). CS scar myometrial thickness was statistically smaller in the niche participant group (RMT over the niche) than in the non-niche (myometrial layer thickness at internal cervical os) group at the first-trimester scan. During the second trimester, myometrial scar thickness (one caliper at the interface of the bladder wall and the myometrium—includes only hypoechogenic layer) was smaller in the non-niche participant group. Still, in the third trimester, there were no differences (Table 3). The median CS scar myometrial thickness of the whole study population over the trimesters is shown in Figure 4.

CS scar myometrial thickness decreased over time in all three trimesters in the non-niche group (*p* < 0.001), compared to between the first and third trimesters in the niche group (*p* < 0.001). The median LUS myometrial thickness at the CS scar area in both participant groups at the second and third trimesters of pregnancy are shown in Figure 5.

The median difference between measurements in CS scar myometrial thickness between the first and second trimesters was 0.7 (IQR, −2.0–1.7) mm, and 2.4 (IQR, 0.8–3.4) mm (*p* = 0.019) between the second and third trimester of pregnancy in the CS scar niche group. To compare with the non-niche group, the median difference in myometrial CS scar thickness between the first and second trimester was 9.4 (IQR, 7.7–12.05) mm (*p* < 0.001), and 1.1 (IQR, 0.2–2.6) mm between the second and third trimester. Still, the change in scar myometrial thickness as percentage between the second and third trimesters was statistically significant in the CS scar niche group compared to the non-niche group (U = 1225; z = −2.438; *p* = 0.015) (Figure 6).

Two cases (4.3%) of uterine dehiscence during the trial of vaginal delivery occurred in the CS scar niche women’s group compared to zero percent in the non-niche group. These two cases underwent emergency CS after trial of labor. The first trimester RMT was 3 mm in the first case and 5.2 mm in the second, and the decrease in myometrial thickness between the first and third trimesters was 0.9 and 3 mm, respectively. This was not statistically different from those of the other participants. In these cases, mean myometrial LUS thickness was 1.6 mm at the second trimester compared to 4.1 mm for the other niche group patients, but did not differ in the third-trimester myometrial LUS thickness (2.1 mm). We observed no uterine rupture cases in the study population.

## 4. Discussion

This study shows that CS scar myometrial thickness is not static and changes throughout pregnancy in the study population. Our data also suggest that in a women’s group with a CS niche in the first trimester, the CS scar myometrial LUS thickness reduces more rapidly between the second and third trimesters. Overall, we have observed that the CS scar niche in the first trimester can possibly predict a significant decrease in the third-trimester LUS myometrial thickness. The relationship between CS scar niche and changes in myometrial scar thickness between the second and third trimesters has not been previously assessed. According to published data, first-trimester CS scar evaluation could be a valuable tool for recognizing high-risk patients in a subsequent pregnancy. Still, previous investigators did not measure scar thickness in the third trimester of pregnancy [4]. On the other hand, another study showed that RMT at the first-trimester scan could not predict the risk of uterine rupture or dehiscence during vaginal birth after CS. The authors observed an absence of correlation between RMT and LUS thickness in the third trimester [8]. Vikhareva Osser et al. observed an association between uterine scar defect before pregnancy and scar defect at delivery [9]. However, they reported pregnancy outcomes in 69 women who underwent transvaginal ultrasonography before pregnancy. The small number of participants and relatively high number of uterine rupture (2.9%) suggest more accurate validation of their findings. Still, study results did not indicate that RMT could be a predictor for uterine rupture. It was reported previously that low uterine segment thickness was a strong predictor for uterine scar defect in a subsequent pregnancy [5,7]. Yet, investigators could not recommend the best cut-off values for use in clinical practice [10]. Gotoh et al. measured the LUS thickness in women’s groups without uterine scar after previous CS [11]. The study found that LUS thickness was the same between women’s groups at the second trimester. It was significantly thinner in women after previous CS from 27 to 39 weeks. These results are consistent with our study because LUS thickness decreases more significantly between the second and third trimester in a women’s group with CS scar niche at the first-trimester scan. To the best of our knowledge, only one previous study described the changes in the dimensions of the uterine scar during pregnancy [3]. The authors concluded that CS scar dimensions change throughout the pregnancy. Moreover, they found that uterine scar defects during delivery were associated with smaller RMT and a more significant decrease of myometrial layer during pregnancy. These findings are similar to our results. We examined that the uterine scar niche at the first trimester has a possible influence on the myometrial LUS thickness decrease between the second and third trimesters. Our observations suggest that changes in uterine scar myometrial thickness are essential in a CS scar niche women’s group. Still, our prospective longitudinal study was not powered to answer the question about a possible association between LUS thickness and uterine rupture or dehiscence. Moreover, there were only two cases of uterine scar dehiscence in our study group of women who underwent vaginal birth, and both cases were in the CS scar niche group. Changes in the myometrial layer were not different from other participants and a single RMT evaluation cannot predict uterine rupture in a subsequent vaginal delivery. In our study population we did not find a relationship between multiple CS and scar niche appearance at first trimester scan. This discrepancy may be due to a small number of participants after previous two and more CS. A cohort study of a large population of women with a history of multiple CS is needed to provide accurate incidence of CS scar niches.

Presently, sonographic evaluation of the CS scar is not usually incorporated in decision making for a mode of delivery, as there is a lack of standard guidelines for patients after one previous CS to attempt vaginal delivery in a subsequent pregnancy [12]. LUS thickness can be measured using a transabdominal and transvaginal probe, and many investigators suggest that a combination of both examinations is the best way to measure LUS thickness [7,13,14]. Moreover, myometrial thickness measurements have been related to the feasible risk of uterine rupture [5,7,11,15]. The presented study investigated changes, not cut-off values, of full LUS and LUS myometrial thickness, and all measurements were taken using TVS. Still, there is no known potential role of the ultrasound assessment of myometrial thickness in managing patients after cesarean delivery. We have previously shown a good inter- and intra-observer agreement for measurements of CS scar niche, RMT, and LUS thickness in all three trimesters of pregnancy [16]. More extensive and good quality studies are indicated to examine the uterine scar myometrial thickness ratio between trimesters or cut-off values for use in clinical practice, and to incorporate these measurements in deciding the mode of delivery or predict complications. Changes in uterine scar thickness throughout pregnancy may suggest the feasibility of uterine scar dehiscence in a subsequent trial of vaginal delivery. The strengths of this study are the prospective and longitudinal design, and that the same researcher took all sonographic measurements, and all were performed according to standardized procedures. The limitations of the presented study are a relatively small sample size of participants and a small number of uterine scar dehiscence during the trial of labour. There was missing data in five cases about the second or third trimester LUS measurements. Therefore, we could not study the influence of the different time intervals from the CS to the subsequent pregnancy and LUS myometrial thickness changes.

## 5. Conclusions

The presented study shows that CS scar myometrial layer changes during pregnancy and thickness in a CS niche group decreases more significantly between the second and third trimesters. Our findings should not lead to changes in clinical practice, but instead justify the need for large and good quality prospective studies to determine the association between scar niche at the first trimester and alteration in LUS thickness in the later pregnancy.

## Figures and Tables

**Figure 1 medicina-58-00407-f001:**
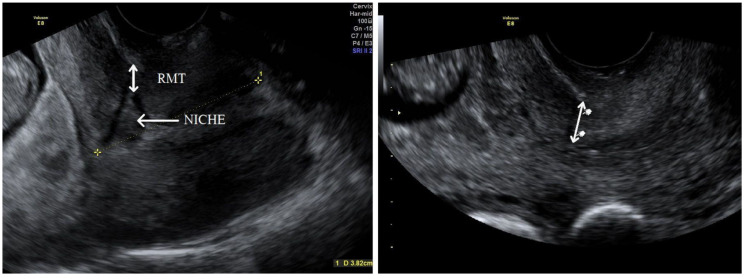
First-trimester midsagittal ultrasound images showing the evaluation of residual myometrial thickness (RMT) (on the **left**) in the CS scar niche participant group and evaluation of myometrial thickness at internal cervical os (on the **right**) in the non-niche participant group.

**Figure 2 medicina-58-00407-f002:**
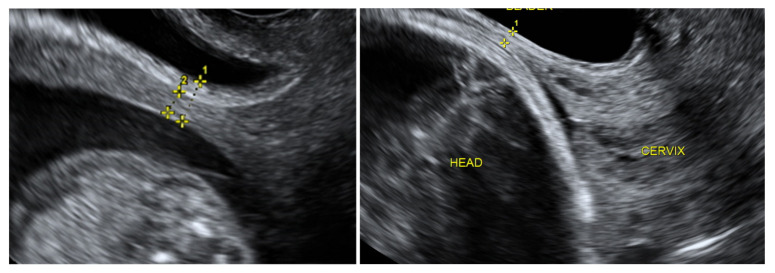
The full low uterine segment thickness (1) (one caliper is placed at the interface between urine and the bladder wall, another is placed at the interface between amniotic fluid (or fetal scalp) and the deciduas); Myometrial LUS thickness (2) (one caliper at the interface of the bladder wall and the myometrium, so includes only hypoechogenic layer) at the second trimester of pregnancy (on the **left**); and full LUS thickness (1) at the third trimester of pregnancy (on the **right**), measured by transvaginal ultrasound.

**Figure 3 medicina-58-00407-f003:**
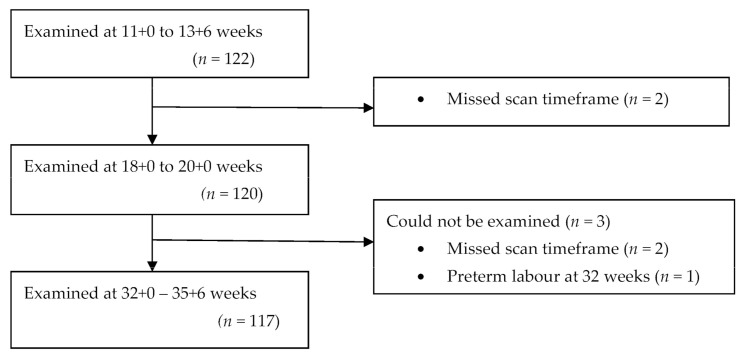
Flowchart of participants through the study period.

**Figure 4 medicina-58-00407-f004:**
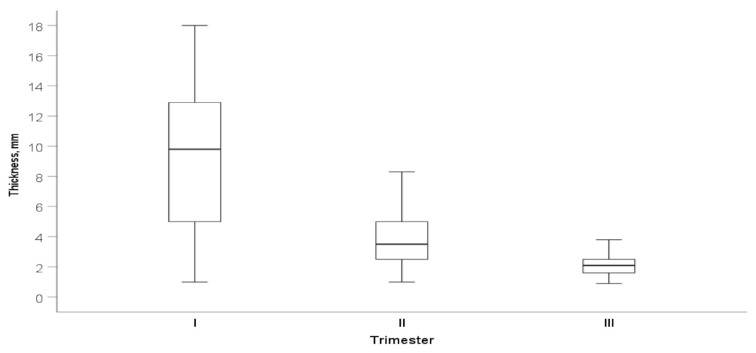
Median scar myometrial thickness (mm) according to trimester of pregnancy. Data are presented as median (IQR).

**Figure 5 medicina-58-00407-f005:**
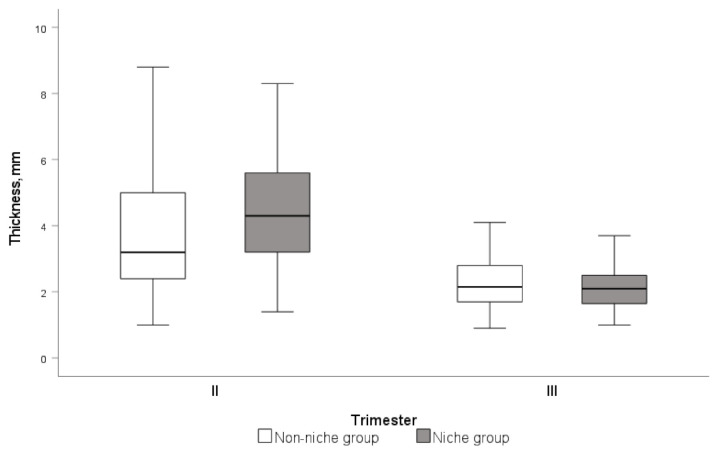
Median myometrial LUS thickness at the second and third trimester of pregnancy in the niche group and non-niche group. Data are presented as median (IQR).

**Figure 6 medicina-58-00407-f006:**
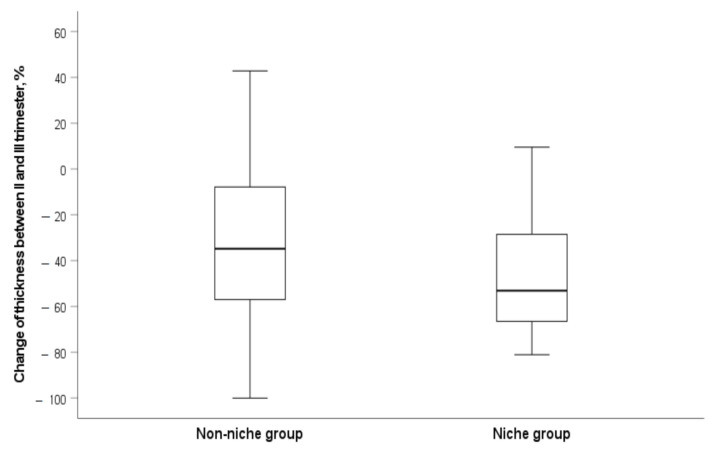
Change in myometrial LUS thickness between the second and third trimester of pregnancy in the CS scar niche group and non-niche group. Data shown as percentage (the more negative percentage, the more significant decrease in myometrial LUS thickness).

**Table 1 medicina-58-00407-t001:** Obstetric outcomes of the study population (*n* = 122).

Parameter	*n* (%)
Successful VBAC	42 (34.4)
Elective repeat CS	48 (39)
Emergency repeat CS	32 (26)
Failed VBAC	22 (34.9)
Vacuum delivery	1 (0.8)
Preterm delivery (before 37 weeks)	7 (5.7)
Induction of labour	27 (42.8)
Use of mechanical dilatation	12 (19)
Oxytocin augmentation	10 (15.8)
Placenta accreta	2 (1.6)
Placenta previa percreta	1 (0.8)
Massive blood loss (≥1500 mL)	2 (1.6)
Transfusion of packed red blood cells	1 (0.8)
Intensive care unit admission	1 (0.8)
Uterine scar dehiscence during CS	2 (1.6)
Chorionamnionitis	10 (15.8)
Adhaesions in pelvis during CS	12 (15)

VBAC, vaginal birth after cesarean section; CS, cesarean section.

**Table 2 medicina-58-00407-t002:** Demographic and obstetric characteristics of the eligible participants.

Parameter	CS Scar Niche Median (IQR) or *n* (%)	Without CS Scar Niche Median (IQR) or *n* (%)	*p* Value
Maternal age (years)	34 (27.9–36.0)	35 (26.0–37.0)	0.486
BMI (kg/m^2^)	24.9 (21.6–28.5)	25.5 (21.9–28.5)	0.529
Gestational diabetes	6 (12.2)	9 (12.3)	1.000
Smoker	5 (10.4)	11 (14.9)	0.662
Previous VBAC	3 (6.25)	5 (6.8)	1.000
Uterine curettage	18 (56.3)	31 (34.4)	0.049
Gestational age (weeks)	39.0 (37.0–39.8)	38.8 (37.1–39.9)	0.525
Birth weight (g)	3589.0 (3183.7–3922.0)	3515.0 (3260.0–3685.0)	0.340
Apgar score after 5 min	10 (9–10)	10 (9–10)	0.951

BMI, body mass index at first trimester; VBAC, vaginal birth after cesarean section; IQR, interquartile range.

**Table 3 medicina-58-00407-t003:** Uterine CS scar thickness during pregnancy trimesters in the niche and non-niche participant groups.

Scar Characteristic	Niche Group (*n* = 49) Median (IQR) mm	Non-Niche Group (*n* = 73) Median (IQR) mm	*p*-Value
Myometrial thickness at the first trimester (presented as RMT in the niche group and thickness near internal cervical oss in the non-niche group)	4.7 (3.3–5.9)	12.8 (11.1–14.9)	0.001
Full LUS thickness at second trimester	7.6 (5.38–9.6)	6.2 (5.1–9.0)	0.138
Myometrial LUS thickness at the second trimester	4.3 (3.2–5.8)	3.1 (2.3–5.0)	0.034
Full LUS thickness at the third trimester	4.0 (3.1–4.5)	4.1 (3.2–5.0)	0.584
Myometrial LUS thickness at the third trimester	2.1 (1.6–2.5)	2.2 (1.7–2.8)	0.379

LUS, low uterine segment; IQR, interquartile range; Full LUS thickness, including tissue between the fetal head and the bladder.

## Data Availability

The data presented in this study are available on request from the corresponding author.

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
