# Peer review of "Cesarean Scar Thickness Decreases during Pregnancy: A Prospective Longitudinal Study"

_medicina, 2022, doi:10.3390/medicina58030407_

Round 1

Reviewer 1 Report

  1. In general, the design lacks apparent clinical significance.
  2. The way the results are presented is cumbersome. 

Author Response

Response to Reviewer 1 comments

Thank you for your comment.

  1. In general, the design lacks apparent clinical significance.

Thank You for your opinion, but we disagree that our study lacks clinical significance.

Safe vaginal delivery after previous caesarean section is a challenge for a patient and a doctor. Decreased ultrasonographic lower segment myometrial thickness was reported as factor associated with an increased risk of uterine rupture in women undergoing vaginal birth after previous caesarean delivery (https://www.rcog.org.uk/globalassets/documents/guidelines/gtg_45.pdf)

Clinical applicability of antenatal LUS measurements in the prediction of a uterine defect in women undergoing vaginal birth after previous caesarean delivery needs be assessed in prospective observational studies (2015) (https://www.rcog.org.uk/globalassets/documents/guidelines/gtg_45.pdf). Preoperative detection of uterine scar dehiscence in women with previous caesarean delivery helps prevent maternal and neonatal morbidity and mortality (Ann Med, 2021;53(1):1265-1269).

We agree, that our limitation of the study is quite small number of participants, but still our results shows threes some evidence to suggest a likely association between CS scar niche in the first trimester of pregnancy and low uterine segment thickness at third trimester of pregnancy as this is improve CS scar integrity at subsequent delivery, of course larger studies are necessary. So, the future studies will require a larger number of participants since rupture of uterine scar is a rare event. However an ultrasound scan at 11 - 14 weeks in the subsequent pregnancy gives support at an early stage of pregnancy and reassurance to women who consider vaginal delivery.

We agree, that our findings should not lead to changes in clinical practice but they justify the need for an appropriately powered prospective study to determine the association between CS scar niche and uterine rupture or dehiscence at a subsequent delivery.

Our study design is novel approach to show how CS scar at first trimester of pregnancy could influence the integrity of the CS scar during pregnancy. We calculated the change in low uterine segment myometrial thickness from the second to third trimester of pregnancy. We think it will be a stimulus for further studies to investigate of sonographic CS scar parameters during pregnancy and to relate to pregnancy outcomes and complications.

  1. The way the results are presented is cumbersome.

Thank You for your opinion, but we disagree that results are presented is cumbersome.

Statistics were consulted on how best to display the study results.

In our study we consistently presented data about the CS scar measurements during pregnancy and the change of the scar thickness between the second and third trimester in two participants groups – with CS scar niche and non-niche. We presented the demographic characteristic of the participants in these two groups, then we showed the measurements of the CS scar niche and low uterine segment thickness at second and third trimesters. Then we showed the results in graphic figures of how changes the low uterine segment myometrial thickness during pregnancy.

Reviewer 2 Report

The authors have made considerable efforts to address my comments on this manuscript. There are still some changes that should be made:

-I believe there is a mistake in Figure 3. The first box is n=117 then n=122. Please check.

-The authors must add in the limitations of the study that they have not collected data about the time interval between the previous CS and the current pregnancy.

-I agree that the authors are not making any conclusion based on: “CS scar niche was visible in 83.7% of women after one previous CS and 16.3% cases after two or more previous CS (p = 0.228)”. However, they must mention in the discussion a possible explanation for these results.

-LUS thickness in the second trimester was different in cases of 1 previous CD vs. 2 or more previous CD. However, LUS thickness in the third trimester was similar among these groups. Based on this, the change between the second and the third trimester might be different in

cases of 1 previous CD vs. 2 or more previous CD. The authors must check this as a subanalysis of the study and include it in the manuscript or as a supplementary.

Author Response

Response to Reviewer 2 comments

The authors have made considerable efforts to address my comments on this manuscript. There are still some changes that should be made:

  • I believe there is a mistake in Figure 3. The first box is n=117 then n=122. Please check.

Thank you for your comment.

Agree. Corrected.

  • The authors must add in the limitations of the study that they have not collected data about the time interval between the previous CS and the current pregnancy.

Thank you for your suggestion.

In our population group, all participants were after at least twelve months after a previous cesarean section. This information was mentioned in the study methods, but we didn’t calculate how many more months was after their last cesarean section. We add it in a discussion as a limitation of the study.

  • I agree that the authors are not making any conclusion based on: “CS scar niche was visible in 83.7% of women after one previous CS and 16.3% cases after two or more previous CS (p = 0.228)”. However, they must mention in the discussion a possible explanation for these results.

Thank you for your comment, we agree this should be mentioned in the discussion.

In previous investigations, authors found that the more CS the higher the prevalence of deficient scars. This could be explained because healing conditions are likely to be poorer in the tissue where there is already a scar. Our result is the opposite, and the discrepancy might be explained by population differences and a relatively small number of participants after one and after more previous CS. The possible explanation is that during the second cesarean section the uterine wall is sutured more precise and with double layer sutures. The method of wound closure could not be examined in all cases, so this is only our suggestion. We add this in discussion.

  • LUS thickness in the second trimester was different in cases of 1 previous CD vs. 2 or more previous CD. However, LUS thickness in the third trimester was similar among these groups. Based on this, the change between the second and the third trimester might be different in cases of 1 previous CD vs. 2 or more previous CD. The authors must check this as a subanalysis of the study and include it in the manuscript or as a supplementary.

Thank you for an important suggestion, we agree that subgroups should be separated. One group after one CS and the second group after two previous CS.

We calculated the results of these two groups separated from those after three previous CS. The statistical analysis in this situation is not changing and the result is the same from a statistical point of view.

 In our study population, there were only two cases after 3 previous CS. So, we grouped these cases into two participants groups – after one CS and after two and more previous CS. According to previous studies we found the analogic situation when results and statistical analysis were performed for the CS scar niche measurements in two groups - after one CS and after two and more CS.

1. Cesarean scar defect: correlation between Cesarean section number, defect size, clinical symptoms and uterine position

 C.B Wang, W.W.C. Chiu. , C.Y. Lee, Y.L. Sun, Y.H.Lin) Ultrasound Obstet Gynecol 2009;34:85-89)

2. Changes in Cesarean section scar dimensions during pregnancy: a prospective longitudinal study

O.Naji, A.Daemen, A. Smith, Y. Abdallah et all. Ultrasound Obstet Gynecol 2013; 41:556-562)

Reviewer 3 Report

  • ,,Five (4.0%) participants had a first-trimester RMT below 2.0 mm, two after the last two and three after the previous one CS.’’ – page 4

The first US during your study was performed between 11th – 13th week. CS scar myometrial thickness less than 2 mm measured in the early pregnancy is one of the diagnostic criteria for caesarean scar pregnancy, proposed by Timor-Tritsch et al.

(Timor-Tritsch, I.E.; Khatib, N.; Monteagudo, A.; Ramos, J.; Berg, R.; Kovács, S. Cesarean scar pregnancies: Experience of 60 cases. J. Ultrasound Med. 2015, 34, 601–610. ) .

There were no such cases reported in your patients. It would be interesting if  the first US examination were performed at early pregnancy (5th – 6th week) to locate the gestational sac precisely. This is just a comment.

If it is possible, please report the following::

  1. How many cases with placenta accrete spectrum disorder were registered among your patients?

    2. How many cases with placenta praevia were registered among your patients?

    3. How many cases with ”silent complete dehiscence” (if any) of Caesarean scar have you noticed during the surgery?

Please report, if possible, what was the method of the uterine closure during the previous caesarean section, single or double layer? Are there differences in the incidence of niche between these two methods of uterine closure?

Explanation: There was evidence showing that there was no difference in any outcomes when comparing single and double layer closure of the uterus according to NICE guideline (NG  192) 2021. (www.nice.org.uk/guidance/ng192.) . Nevertheless, there are no such studies on caesarean scar thickness dynamics during the pregnancy.

Author Response

Response to Reviewer 3 comments

Thank you for your comments and questions, we find them very interesting and useful to improve our manuscript.

  1. The first US during your study was performed between 11th – 13th week. CS scar myometrial thickness less than 2 mm measured in the early pregnancy is one of the diagnostic criteria for caesarean scar pregnancy, proposed by Timor-Tritsch et al. (Timor-Tritsch, I.E.; Khatib, N.; Monteagudo, A.; Ramos, J.; Berg, R.; Kovács, S. Cesarean scar pregnancies: Experience of 60 cases. J. Ultrasound Med. 2015, 34, 601–610. ) . There were no such cases reported in your patients. It would be interesting if the first US examination were performed at early pregnancy (5th – 6th week) to locate the gestational sac precisely. This is just a comment.

In our study we performed sonographic evaluation of previous CS scar from 11 weeks of gestation and there were no any cases of CS scar pregnancy. We agree, it would be interesting to check the scar at 5-9 weeks of gestation, but our methodology was only from the 11 weeks of gestation when women are coming for prenatal ultrasound screening.

  1. How many cases with placenta accrete spectrum disorder were registered among your patients?

In our study population there were only three cases of PAS - 2 cases of placenta accreta after previous one cesarean delivery, they went for vaginal delivery in a subsequent pregnancy (blood loos 2000ml in both cases) and 1 case of placenta percreta after previous two cesarean deliveries, she went for planned CS at 34 weeks of gestation because of placenta previa percreta (in this case ulrasonographic markers were seen from the first trimester) and blood loos was 2500ml.

  1. How many cases with placenta praevia were registered among your patients?

There were only one case of placenta previa those with placenta percreta.

  1. How many cases with ”silent complete dehiscence” (if any) of Caesarean scar have you noticed during the surgery?

There was no silent dehiscence during the surgery after one or two and more previous CS, in one case for emergency CS for suspected scar rupture they found only thin muscular layer but no dehiscence or rupture.

  1. Please report, if possible, what was the method of the uterine closure during the previous caesarean section, single or double layer? Are there differences in the incidence of niche between these two methods of uterine closure?

 Explanation: There was evidence showing that there was no difference in any outcomes when comparing single and double layer closure of the uterus according to NICE guideline (NG 192) 2021. (www.nice.org.uk/guidance/ng192.) . Nevertheless, there are no such studies on caesarean scar thickness dynamics during the pregnancy.

In our study we cannot collect such information about uterine closure, the technique differs between hospitals, but in Lithuania usually we use single layer closure.

Reviewer 4 Report

Thank you for the opportunity to review your work. Below you can find some aspects I suggest should be clarified. 

In the Abstract section: 

"the full LUS and myometrial LUS thickness at 18 – 20 and 32 – 35 weeks of gestation were measured in the thinnest part of the scar" -please explain what does it mean the full LUS and myometrial LUS thickness?? Are 2 different measurements ?? Or is it the same?

 "Myometrial LUS thickness as percentage decreases significantly between the second and third trimester in the CS scar niche group compared with a non-niche group (U = 1225; z = -2.438; p = 0.015)" - Why did you need to make a % comparison? Please explain

Material and Method section

"LUS is a two-layer structure with a hyperechogenic layer, including the bladder wall and a hypoechogenic layer that is considered to represent the myometrium [5] (Figure 2). At least three measurements were taken, with the lowest value being retained" - Where was the measurement performed? You mentioned three measurements but what exactly are the landmarks? Please be more precise.

"These characteristics were compared between first, second, and third trimesters using two-way Friedman’s ANOVA test" - Where am I performing the measurements in the second and third trimester? Please be more specific

Results section

"The mean gestational age in a study population at delivery was 38.8±2.37 weeks with a neonatal weight of 3473.7±598.0 grams" - please add additional information about the mode of delivery: CS or VBAC?

"CS scar myometrial thickness was statistically smaller in a participants niche group (RMT over the niche) than in a non-niche (myometrial layer thickness at internal cervical os) group at the first-trimester scan" - please mention where the corresponding data can be found in the text. 

Author Response

Response to Reviewer 4 comments

Thank you for the opportunity to review your work. Below you can find some aspects I suggest should be clarified. 

  • "the full LUS and myometrial LUS thickness at 18 – 20 and 32 – 35 weeks of gestation were measured in the thinnest part of the scar" -please explain what does it mean the full LUS and myometrial LUS thickness?? Are two different measurements ?? Or is it the same?

Thank you for your comment.

The methods of measurements during the second and third trimester of pregnancy are explained in the methods, and in figure 2, Full LUS thickness and myometrial thickness is not the exact measurements. The methods of LUS measurements are described in the methods.

“TVS on a full bladder was used to measure the full LUS and myometrial thickness to delineate the scar area clearly, as was previously described [5] (Figure 2). LUS is a two-layer structure with a hyperechogenic layer, including the bladder wall and a hypoechogenic layer that represents the myometrium [5] (Figure 2).

  • "Myometrial LUS thickness as percentage decreases significantly between the second and third trimester in the CS scar niche group compared with a non-niche group (U = 1225; z = -2.438; = 0.015)" - Why did you need to make a % comparison? Please explain

As myometrial thickness in CS scar with niche was already different from non-niche patients at the first and second trimesters. Hence, we think this change should be calculated as a percentage of the initial thickness, not as an absolute number.

  • "LUS is a two-layer structure with a hyperechogenic layer, including the bladder wall and a hypoechogenic layer that is considered to represent the myometrium [5] (Figure 2). At least three measurements were taken, with the lowest value being retained" - Where was the measurement performed? You mentioned three measurements but what exactly are the landmarks? Please be more precise.

Agree, corrected and added to the methods of measurements. They were written after ultrasound pictures of LUS and Myometrial LUS thickness measurements.

To measure the full LUS thickness on a full patient’s bladder – one calliper is placed at the interface between urine and the bladder wall. Another is placed at the interface between amniotic fluid or fetal scalp and the decidua. The myometrial thickness is measured with the one calliper at the interface of the bladder wall and the myometrium to include only hypoechogenic layer.

  • "These characteristics were compared between first, second, and third trimesters using two-way Friedman’s ANOVA test" - Where am I performing the measurements in the second and third trimester? Please be more specific

For paired samples to compare LUS myometrial thickness at second and third trimester in a CS scar niche group and non-niche group, we used the Wilcoxon Signed Rank test. It is described in methods.

  • "The mean gestational age in a study population at delivery was 38.8±2.37 weeks with a neonatal weight of 3473.7±598.0 grams" - please add additional information about the mode of delivery: CS or VBAC?

Corrected added information about mode of delivery in the results.

  • "CS scar myometrial thickness was statistically smaller in a participants niche group (RMT over the niche) than in a non-niche (myometrial layer thickness at internal cervical os) group at the first-trimester scan" - please mention where the corresponding data can be found in the text. 

Corrected, added mention in the text.

Reviewer 5 Report

I think there could be an error in figure 3. Flowchart of participants through the study period
Perhaps it is not appropriate to place the first 2 frames in this place (those where it indicates "Examined at 32 + 0 to 35 + 6 weeks
(n = 117)" and "Could not be examined (n = 2) Missed scan timeframe (n = 2)")
since they are repeated below
The words "Changes in" should be deleted from the first paragraph of page 6

It would be interesting, if the data were available, to know the correlation between the onset of labor or the mode of elective termination of pregnancy in each group studied, as well as in the cases in which caesarean section scar dehiscence occurred

Author Response

Response to Reviewer 5 comments

  • I think there could be an error in figure 3. Flowchart of participants through the study period

Thank you for your comment.

Corrected

.
                Perhaps it is not appropriate to place the first two frames in this place (those where it indicates "Examined at 32 + 0 to 35 + 6 weeks
(n = 117)" and "Could not be examined (n = 2) Missed scan timeframe (n = 2)") since they are repeated below

Thank you for an important correction.

Corrected figure 3

                - The words "Changes in" should be deleted from the first paragraph of page 6

Thank you

Corrected, deleted

                - It would be interesting, if the data were available, to know the correlation between the onset of labor or the mode of elective termination of pregnancy in each group studied, as well as in the cases in which caesarean section scar dehiscence occurred

Information about mode of elective termination of pregnancy and all types of delivery was described in our previous manuscript, but we added some information in a results.(Reference 16)

”Of the 63 women who underwent a trial of labor after one previous CS, 41 (65.0%) had a successful vaginal delivery. Vacuum extraction was performed in one case (1.5%) among these women. Of 27 (42.8%) women with labor induction after one previous CS, 15 (55.5%) had a vaginal delivery. Emergency repeat CS after trial of labor was performed in 23 (36.5%) women, 10 due to non-reassuring fetal status and 13 due to arrest of labor. In the patient group with visible CS scars at the first trimester of pregnancy, seven delivered through natural pathways, as did the same number of women in the group with non-visible CS scars (55.7% vs. 58.3%, p = 1.000). In the patient group with CS scar niches (n = 49) for comparison with the non-niche group (n = 46), there were no statistical differences in the type of delivery. A total of 19 women had successful trials of labor in the niche group and 22 in the non-niche group (38.7% vs. 47.8%, p = 0.802). Fifteen underwent elective repeat Cesarean delivery for various clinical reasons in the niche group, in comparison with 33 women in the non-niche group (31.9% vs. 44.6% p = 0.337). Thirteen women required intrapartum emergency CS because of failed trials of labor in the niche group, versus 19 women in the non-niche group (40.6% vs. 46.3% p = 0.802). The mean gestational age of the study population at delivery was 38.8 ± 2.37 weeks, with a neonatal weight of 3473.7 ± 598.0 g. The median neonatal weight did not differ between the groups according to the CS scar niche and the Apgar score. Median birth weight in the CS scar niche group was 3589.0 (IQR 3183.7–3922.0) g, compared with 3515.0 (IQR 3260.0-3685.0) g in the non-niche women’s group (p = 0.340). After 5 min in both groups, the median Apgar score was 10.0 (IQR 9.0–10.0) (p = 0.951). There were two (4.3%) cases with a neonatal Apgar score less than 7 after one minute in the CS scar niche group, compared with a single case (1.4%) in the non-niche women’s group. Two women (4.3%) had uterine dehiscence confirmed following a trial of vaginal delivery, and both had CS niches during the first trimester. No uterine ruptures occurred in the study population.”

Savukyne E, Machtejeviene E, Paskauskas S, Ramoniene G, Nadisauskiene RJ. Transvaginal sonographic evaluation of cesarean scar niche in pregnancy: a prospective longitudinal study. Medicina 2021;57(10) 1091.

This manuscript is a resubmission of an earlier submission. The following is a list of the peer review reports and author responses from that submission.

Round 1

Reviewer 1 Report

In the paper, Savukyne et al. conducted a prospective longitudinal study recruiting 122 patients and measured the CS thickness in the first, second and third trimesters, a remark decrease in scar decrease in women with CS niches as compared with the non-niche group. They suggested that the CS myometrial layer thickness would be considered as an index for assessing the potential risk of pregnancy complications in women with previous CS, especially in patients has CS niches.

This manuscript seems to be relevant to the field. However, several concerns should be considered:

Major Comments:

General:

  1. The criteria of recruitment in this study are unclear.
  2. According to Abstract and Table 2, three terms were used for describing the thickness of CS in the manuscript: ‘CS thickness’, ‘Full LUS thickness’, ‘Myometrial thickness’. However, the difference between these terms is unclear in the Results.
  3. ‘CS’ stands for ‘Cesarean scar (CS)’ or ‘Cesarean section (CS)’?

Specific:

  1. Line 53-59: in the criteria of the study recruitment, pregnancy in the CS hasn’t been mentioned. Any indexes concerning the diagnosis of CSP or CS scar niche should also be considered?
  2. Line 91: double blank space before the sentence.
  3. Line 120-121 & 123-125: The interpretation of Table 2 and Fig 3 are unmatched for the data shown. Does it seem that a more significant decline of CS thickness would be found in the Non-niche group instead of the Niche group?
  4. It is suggested that discussion concerning CSP and CS niches found before current pregnancy may be needed.
  5. Line 180: double blank space before LUS.
  6. Data/Results shown previously may not match with the Conclusion?

Author Response

Dear reviewer,

Thank you for all your questions and suggestions. We found them very important and useful to improve our manuscript.

Hope you will find our answers clear and useful.

Best wishes

Reviewer 2 Report

Although the data in this article are clearly presented and very user friendly, i do not think this article brings any significant novelty to the field.

Author Response

Dear reviewer,

Thank you for your comment about our manuscript.

We have shown that CS scar myometrial thickness measured with TV probe is not static and change over the course of pregnancy. We have observed that the change in CS scar myometrial thickness is different according to CS scar niche and non-niche women's group. To date, the literature has focused on the appearance of CS scars on single assessments of scar thickness or morphology. Our study shows for the first time that CS scar niche may have a potential role in the late second and third trimester LUS myometrial thickness.

From a practical clinical point of view, this study was designed to fit with routine clinical management - first-trimester screening, fetal biometry and anomaly screening in the second trimester and scans for the fetal growth assessment in the third trimester.

Hope you found the answer useful.

Best wishes

Reviewer 3 Report

This study investigated the myometrial thickness in patients with previous cesarean section(s) in the three trimesters of pregnancy. The main strength of the study is its prospective design. However, many aspects of the methods and results should be improved. The conclusions are not fully based on the results of the study.

My comments are as follows:

Major comments:

-The authors must use statistical tests for paired samples.

-Were there any differences in baseline characteristics between niche and non-niche?

-Were there any differences in the myometrial thickness (in any trimester) according to the time interval from the previous CS and the current pregnancy?

-Were there any differences in the myometrial thickness (in any trimester) according to having a successful VBAC?

-Did the authors have any data on changes on myometrial thickness in patients with no previous CS?

-Lines 89-90: “CS scar niche was visible in 83.7% of women after one previous CS and 16.3% cases after two or more previous CS (p = 0.228).” Could the authors comment on this? Why do they think the niche was seen more in patients with only one previous CS section?

-Could the results be different if they included only patients with one previous CS?

-Lines 137-138 and 191-192: “Our data also suggest that in a women’s group with a CS niche in the first trimester, the myometrial thickness reduces more rapidly between the second and third trimesters.” This conclusion is not correct. Myometrial thickness in CS scar with niche was already different from non-niche patients at the first and second trimesters so this change should be calculated as a percentage of the initial thickness not as an absolute number.

-Figure 3 is misleading because the measurement in the first trimester is not the same as the second and third trimesters and could not be compared with them (I suggest to remove the first trimester measurement from the figure).

-Have the authors looked at the correlation between the measurements in the 3 trimesters on an individual level not as a group?

-Were the measurements obtained in all the participants in the 3 trimesters? Are there any missing data?

-Lines 176-177: “LUS thickness can be measured using transabdominal and transvaginal probe, and many investigators suggest that a combination of both examinations is the best way to measure LUS thickness” based on this why did authors use only TVS?

-The conclusion is not based on data from the current study.

Minor comments:

-There is a typo error in Lines 43 and 70: SC should be CS.

-There is a typo error in the figures: thikness should be thickness.

-Line 103: “CS scar myometrial thickness was statistically smaller…” It’s more precise to say: “changes in CS scar…”

-There is a typo error in Lines 166: egzaminated should be examinated.

-Line 187: “and that the same operator took all sonographic measurements” this has not been mentioned in the methods. Who was this operator?

Author Response

Dear reviewer,

Thank you a lot for your comments and suggestions according to our manuscript. We found them very useful to improve our manuscript. Hope you will find them clear and correct.

Best wishes

Round 2

Reviewer 3 Report

The authors have not made sufficient changes in the manuscript according to the points raised during the first round of revision. Statistical methods and Figure 3 must be revised. Data that were not collected b the authors that may affect the results must be mentioned in the limitations section.

Author Response

Thank you for a careful reading of our text and suggestions.

All comments we received on this study have been taken into account in improving the quality of the article, and we present our reply to each of them separately.

Point 1:Statistical methods

Point 2:and Figure 3 must be revised

Point 3:Data that were not collected b the authors that may affect the results must be mentioned in the limitations section.

Response 1:

According to previous comment, we changed a statistical method for a pairwise samples (non-parametric) and used a Wilcoxon Signed-Rank test to compare paired data. We described it in a methods.

Line 17 and Lines 78-79.

Response 2:

According to your comment, we revised Figure 3 and left only measurements of second and third trimester and described that it was median LUS myometrial thickness at second and third trimesters of pregnancy, not a change in a thickness of CS scar.

Line 126-126.

Moreover, we calculated change of the CS scar thickness between the second and third trimester (in two patients group according the niche in the first trimester) as a percentage change  and put a new figure in the text (Figure 4.) The calculations were done as:

Myometrial thickness III minus Myometrial thickness II / Myometrial thickness II x 100

The more negative percentage – the more significant decrease in a group. To compare measurements in two groups were used Mann -Whitney test.

Line 132-133.

Response 3:

According to previous comments we described as limitation of a study in a discussion.

Line 202.

Thank you again for your comments and suggestions.
